# Modern Trends in Neutron Scattering Instrument Technologies

Georg Ehlers [1,*], Morris L. Crow [1], Yacouba Diawara [1], Franz X. Gallmeier [1], Xiaosong Geng [1],
Garrett E. Granroth [2], Raymond D. Gregory [1], Fahima F. Islam [1], Robert O. Knudson IV [1], Fankang Li [1],
Matthew S. Loyd [1] and Bogdan Vacaliuc [1]

[1] Neutron Technologies Division, Oak Ridge National Laboratory, Oak Ridge, TN 37831, USA;
crowmljr@ornl.gov (M.L.C.); diawaray@ornl.gov (Y.D.); gallmeierfz@ornl.gov (F.X.G.); geng@ornl.gov (X.G.);
gregoryrd@ornl.gov (R.D.G.); islamff@ornl.gov (F.F.I.); knudsoniroiv@ornl.gov (R.O.K.IV);
frankli@ornl.gov (F.L.); loydms@ornl.gov (M.S.L.); vacaliucb@ornl.gov (B.V.)
[2] Neutron Scattering Division, Oak Ridge National Laboratory, Oak Ridge, TN 37831, USA;
granrothge@ornl.gov
* Correspondence: ehlersg@ornl.gov

**Abstract:** This article reviews some current trends that can be observed in the development of neutron scattering instrument technologies. While the number of neutron scattering facilities worldwide and the number of beam days they offer are largely stable, their scientific impact is increasing through improving instrumental capabilities, new and more versatile instruments, and more efficient data collection protocols. Neutron beams are becoming smaller but more intense, and instruments are being designed to utilize more 'useful' neutrons in unit time. This article picks and discusses a few recent developments in the areas of integrated source and instrument design, use of computational tools, new detectors, and experiment automation.

**Keywords:** neutron scattering; detector; polarization; virtual instrument





## 1. Introduction

Neutron scattering is an experimental probe for the structure and dynamics in materials on the atomic (molecular) scale. Some of the essential and fundamental contributions that underpin modern technology and society have been made by neutron scattering research [1,2]. Good overviews of the modern science and technology of neutron scattering are offered in the books by Willis and Carlile [3], Carpenter and Loong [4], and Boothroyd [5].

Most of the steady advance in neutron scattering science is directly linked to the continued development of the underlying instrument technologies. The brightest sources (by time-average) in operation today, the high-flux research reactor operated by the Institut Laue-Langevin (ILL) [6] in Grenoble, France, the FRM-II reactor near Garching (Germany) [7], and the high-flux isotope reactor (HFIR) [8] at Oak Ridge National Laboratory (ORNL), USA, have been operating at today's performance levels for many years. The PIK reactor in Russia [9] is expected to reach comparable performance once routine operation at design power commences. In the same time span of roughly 50 years, all instrument types have seen huge performance gains, some by a factor of $\sim 10^3$ or more, depending on the metrics used, which are due to the concurrent development of enabling technologies.

Pulsed neutron sources became operational in the 1970s and 1980s [10]. These sources produce neutrons by spallation of heavy nuclei, instead of fission. The pulsed nature of the beam emerges from a mere convenience to produce a high-energy proton beam that strikes a heavy metal target to produce the neutrons by spallation. Using RF technology, one can accelerate protons more efficiently in pulses than with static electric fields, while keeping the beam focused [11]. However, the pulsed nature of the beam is also essential to support time-of-flight methods critical to using higher energy neutrons ($E \sim 1$ eV) that

cannot not be effectively monochromated with crystal monochromators. As spallation creates less heat per neutron than fission, and neutron production can be compressed in time [12], higher peak pulse brightness can ultimately be realized than the average brightness at a reactor source. Here, brightness is the number of neutrons per unit area, unit time, and unit solid angle emitted from the source into the direction of neutron beam lines. Spallation sources can generate roughly 3–4 times more neutrons per deposited energy in the neutron production zone than reactors. New pulsed neutron sources that have been constructed (ISIS [13], SNS [14], J-PARC [15], and CSNS [16]), or are being constructed (ESS [17] and STS [18]), are actually less bright if one considers the time-average brightness. The peak brightness in a pulse, however, is much higher than at a reactor source. Neutron scattering instruments can be designed specifically to make use of the pulsed structure of the beam. Enormous efficiency gains were realized for some instrument types by going from continuous beam to pulsed beam (such as powder diffraction, for example), where performance scales with source peak brightness and not time-average source brightness.

## 2. Technology Trends

At a high level, one can observe the following trends, some of which are being discussed in this article in more detail.

- Operating neutron sources are becoming more powerful and the facilities become larger, but their overall number slowly decreases with time. This development is rooted in the funding mechanisms of fundamental research by national agencies, who tend to listen to capability arguments more than capacity arguments, in combination with the increasing cost to run these facilities. The decreasing number of operating facilities is a trend the community is actively trying to turn around.
- While scattering instruments exist for the primary purpose of facilitating scattering experiments in a certain science area, one looks at them today more like science laboratories. Samples can often be manipulated in more than one way, even going to non-equilibrium states in pump–probe experiments, while scattering data are taken.
- A general trend is to study smaller samples, with smaller (more focused) and more intense beams.
- New strategies for multiplexing beams are being developed and implemented in order to make use of as many neutrons simultaneously as possible.
- Automate as much as possible. Support a greater diversity of experiments at an instrument. Computational and experiment planning tools are becoming increasingly important.
- Some scattering cross sections depend on the neutron spin direction relative to the scattering vector, which ultimately leads to the truly unique capabilities of neutron scattering that other probes do not offer. In order to make better use of these unique capabilities, technology and measurement protocols are being developed for more sophisticated control and the manipulation of the neutron spin in an instrument.
- The remote control of scattering experiments at neutron beam lines was a capability in planning when its development was accelerated by the COVID-19 pandemic. For various reasons, neutron scattering experiments have traditionally not been conducted remotely. This capability has now been implemented at most beamlines that are in the user program at neutron facilities around the world.

### 2.1. Integrated Design of Sources and Instruments

The modern view is that sources and instruments should be seen as a unit and should preferably be designed in parallel for best instrument performance. About 10 years ago, it was realized that the best neutron optical transport from the source to the sample can be achieved in most cases when the source and sample are approximately of the same size [19]. The more traditional view that larger sources (by volume) are better, because they produce more neutrons, has been overturned by the realization that smaller sources can be made brighter, emitting more neutrons per unit surface area and unit time [20–22]. At the

same time, neutrons produced in the periphery of a larger source are far less likely to be transported to a sample position.

The European Spallation Source (ESS) was the first facility that conducted a concurrent design of a source and instrument suite intentionally and thoroughly [17]. The result of this study was that a much reduced moderator height of about 3 cm was the best overall value for all instruments. This is a factor 3–6 smaller than existing high-performance sources [23–25]. It was found that an even smaller source can be made even brighter, but that one would expect artifacts due to an imperfect neutron transport to counter-balance this gain [17]. The SNS Second Target Station (STS) at ORNL will be the next major facility (after the ESS) to be built and will go through a similar global optimization process [26,27].

Neutron scattering instruments have traditionally always needed relatively large samples (with a mass on the order of ~grams, unless the sample contains hydrogen). This is due to the comparatively low flux offered by even the best sources, and the resulting low count rates. For a number of reasons, the average sample size is slowly getting smaller over time. The biggest scientific impact is in new materials, which can often only be synthesized in small amounts. It is therefore often preferable—if the desired experimental resolution conditions permit—to use a focused small beam at the sample location with higher neutron flux over a smaller area. Studying samples under extreme conditions of pressure, temperature, and magnetic field (separately or simultaneously) will put practical limits on the sample size. Isotopic enrichment is an important strategy in neutron scattering to enhance the signal in a specific situation, again limiting the amount of sample because of the high cost involved.

Great progress has been made over the last decade in the state of the art of computer simulations of neutron beam transport and instrument performance. A few years ago, such a simulation would be complete with an estimate for the neutron flux at the sample position. One would judge the instrument performance by comparison with known installations and the flux they offer. Today, one routinely simulates the scattering from the sample and its environment as well, resulting in much more detailed performance predictions. More detail will be given below in Section 2.6.

Nowadays, the design of all new instruments includes extensive computer-aided performance optimizations. Projects to design and build new, medium-sized, neutron scattering facilities [28], such as the Jülich high-brilliance neutron source project [29,30] or the SONATE project in France [31], take an integrated approach for the design of a source and instrument suite. The smaller scale of these facilities and the lower operating power translate into lower costs for construction and operation, and performance optimizations are critical to achieve nevertheless competitive instrument performance. The need for these facilities is real: many neutron scattering applications need capacity more than premium flux. For example, the development of new technologies, quality assurance of instrument components with beam (neutron guides), routine characterization of components used in industrial processes, and the training of the next generation of scientists all need the beam capacity that such facilities will offer.

An area which is quickly growing and gaining importance is to design custom components such as collimators [32,33] in software and to fabricate them with additive manufacturing [34,35]. This new technology allows one to achieve shapes and geometries otherwise impossible or cost-prohibitive. The computer-aided analysis, nowadays, reaches a much higher level of sophistication than simply simulating signal-to-noise or neutron transmission for a specific situation [33]. The realistic, accurate simulation of sample scattering allows geometrical parameters, that change the collimation, to be refined before anything is built [32]. The materials can also be modified to understand how a collimator that removes neutrons primarily by scattering works differently from one that removes primarily by absorption [33]. Component designs can directly be output in formats suitable for CAD software [36] which expedites the time from design to manufacture. Note that the connection to CAD permits a closer optimization with traditional manufacturing processes as well.

## 2.2. Multiplexed Neutron Beams

Because the source brightness is relatively low, when compared to X-ray sources, a great deal of innovation is spent to design instruments which make use of beams that are multiplexed in one way or another. The goal is to make simultaneous use of as many useful neutrons as possible. The simplest example for this idea is to use a multidetector ('banana' detector) which is a standard technical solution for powder diffraction instruments [37,38], but there are also many other examples [39–42].

Triple-axis instruments with a multiplexed analyzer have been put in operation in various facilities [43–48]. The multi-analyzer arrangement gives a net increase in the data collection rate as long as the resolution is equally good in all simultaneously acquired data. A challenge remains to reduce the parasitic scattering in the secondary spectrometer to a minimum.

For direct geometry inelastic time-of-flight spectrometers, an analogous concept is known as repetition rate multiplication, RRM [49–51]. At a reactor source, this instrument type can freely choose the base frequency at which it runs. At a pulsed source, the instrument is bound to the source frequency which is not necessarily optimized (generally much too slow). Therefore, the concept was developed to make use of several short pulses out of the main pulse. All new instruments of this type are designed and built this way [52–54], and several are in construction [55–57].

## 2.3. Detectors

Thermal neutron detectors are essential components for neutron scattering instruments. The most important characteristics are a high efficiency ($\gtrsim$70%) for thermal neutrons combined with a very low sensitivity to other types of ionizing radiation, such as high-energy neutrons and $\gamma$-radiation. Other desired properties, where the importance varies with instrument type, are high count-rate capability, high position resolution (sub-mm to microns), excellent position linearity and stability, electrical stability, and the ability to operate under high incident thermal neutron flux.

The perhaps most common type of detector relies on the high thermal neutron absorption cross section of the $^3$He isotope. The grand total of $^3$He gas in the present installations at various neutron sources worldwide exceeds 90,000 liters. The current demand for $^3$He gas to use in new or upgraded neutron scattering instruments by far exceeds the available amount on the market [58].

The most significant technical advance of the last decade in thermal neutron detectors is the development of a new type of detector that makes use of charged particles generated in neutron capture by the $^{10}$B isotope [59–63]. The boron is arranged in a thin solid layer of boron carbide $B_4C$ which is readily available and offers great boron density. Gaseous detectors using $BF_3$ have been used for a long time but suffer from various technical difficulties [64]. The operating pressure cannot be as high as with $^3$He, while at the same time, the capture cross section of $^{10}$B is lower. Therefore, the overall efficiency in the thermal range is not competitive. The $BF_3$ gas is also highly toxic which is an operational concern.

The solid-layer converter detector based on boron was developed as a response to the general shortage of $^3$He. It was realized that a viable alternative was needed because, as a non-renewable resource, helium supply would never be able to meet the strongly increased demand that one can anticipate by looking at the upgrade plans at various neutron scattering facilities.

The detector relies on the charged alpha particle generated in the neutron capture to escape the solid boron carbide layer. Therefore, this layer is designed to be thin, ~1 μm. The detection efficiency of a single layer is low, less than 5%, and 10–30 layers are stacked in a closely packed array to bring the overall efficiency into a useful range. Variants of this detector type exist in which the layers are arranged as flat sheets ('multigrid' detector [62]) or as cylindrical tubes ('straw' detector [63]). The gas between the detection layers (typically an $Ar/CO_2$ mixture near 1 atm) and a static electric field are used to operate the detector, very much like a traditional gas detector ('proportional counter').

The biggest challenges to make such a detector competitive are (i) the large, desired area with homogeneous coverage (a time-of-flight inelastic instrument would need $\sim$50 m$^2$) and (ii) to limit the amount of internal secondary scattering in the detector. Any neutron scattered in the detector must be considered lost, as it is not known how to discriminate these neutrons. The biggest promise of this detector type for advance over the traditional helium-based proportional counter is the local rate capability, as demonstrated with the multigrid detector [62].

Thermal neutrons can also be detected by means of neutron absorption in a scintillator material (glass) that contains $^6$Li and Ce$^{3+}$. The neutron absorption creates charged particles with kinetic energy, which interact with the glass to produce ionization. This can excite Ce$^{3+}$ ions out of their ground state, and the photons emitted with the return of the Ce$^{3+}$ ions to their ground state can be multiplied and collected. Each absorbed neutron can generate thousands of photons in the scintillator [65]. Neutron scattering instruments often use scintillator detectors when a relatively small physical detection area is sufficient, or when superior spatial resolution is needed that a gas detector cannot achieve.

A key requirement for a neutron detection system at a neutron scattering instrument is that the detector area is evenly covered, without gaps, and with little variation in the physical characteristics of the detector over the area. A tiled arrangement of flat scintillator screens will necessarily introduce gaps which one strongly desires to avoid. In a new development at ORNL in collaboration with RMD [66], a homogeneous coverage of 2.91 sr (no gaps) was achieved with a neutron-sensitive scintillator surface, see Figure 1. The light is extracted with tapers that guide the light to photomultipliers. These are read out individually and the detector image is rearranged in software as a two-dimensional flat image, see Figure 2.

No neutron detector is perfect, and they remain a primary area of development. For many applications, the most important development priorities are to improve the count rate capability and the spatial resolution [67].

Neutron imaging is a rapidly developing field [68–70] which offers great challenges to the count rate capability of a detector: rather than counting the scattered 5% of a beam, one aims to count the remaining 95% of the beam that is transmitted (not accounting for absorption). As a consequence, a small surface detector will have to manage a count rate exceeding the current capability limits by about two orders of magnitude. Imaging also has higher requirements for the spatial resolution and timing resolution (at a pulsed source).

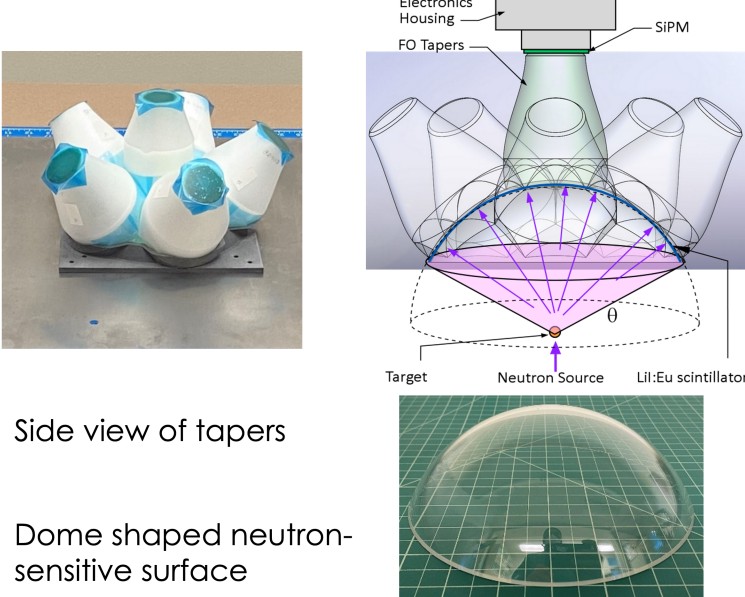

**Figure 1.** Physical make-up of the dome detector.

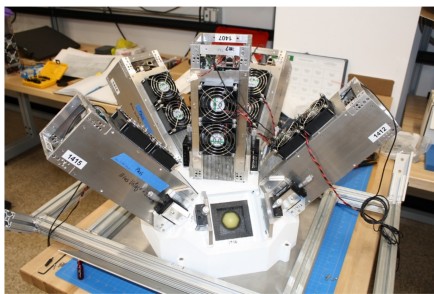
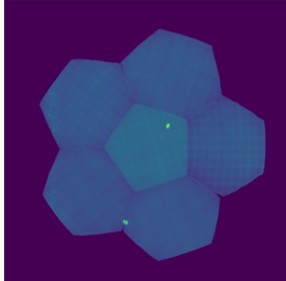

**Figure 2.** Assembled prototype and a pair of Bragg peaks collected with a single crystal and monochromatic beam.

Another area where one can expect more development is in the area of neutron monitors. Neutron beam monitors are detectors with an intentionally very low efficiency. They are important diagnostic tools in the beam transport system of an instrument and can also be used to monitor the source performance of a facility. Some instrument types use monitors for data normalization. The future generation of neutron beam monitors should provide fast time resolution (sub nano-second), very strong $\gamma$-ray rejection, and should be position-sensitive for beam profiling.

### 2.4. Polarized Neutron Beams

The magnetic moment of the neutron enables it to probe magnetic moments due to unpaired electrons in a material. A spin-polarized beam can be used to study magnetic correlations exclusively by direct discrimination from other (nuclear) interactions [71,72]. Similarly, one can use a spin-polarized beam to distinguish between nuclear coherent scattering (measuring pair correlations between atoms or molecules) and nuclear spin-incoherent scattering (measuring self-correlation functions). This is an experimental capability specific to neutron scattering, and the reader is referred to textbooks for more details [3–5]. In this application of spin-polarized beam operation, the beam polarization is used to enhance the sensitivity of an experiment for a specific type of interaction over another.

One can also spin-polarize a beam and then manipulate the beam polarization further, such that Larmor precession of the neutron spin occurs in a magnetic field that is perpendicular to the neutron spin direction. By carefully controlling the magnitude and direction of the field, one can use the neutron spin as a clock and as a measure for the neutron speed or its direction. This measure is much finer than what can be achieved with traditional means through making a beam monochromatic (by using a monochromator crystal or a chopper system), or by using collimation. This is to say that it is possible to achieve higher resolution, as the total precession angle of a neutron can be as high as $\sim 10^4$–$10^5$ rad. Another key point is that one also eliminates the need to remove neutrons from the beam that do not have the desired speed or direction, as one can make difference measurements of the total precession angle on both sides of the sample, in sufficiently homogeneous fields and symmetric situations. As a result, one can enhance the resolution of a host instrument for a specific variable (energy transfer or momentum transfer) without a corresponding intensity ($i$) loss that would roughly scale with resolution ($r$) as $i \propto r^{-4}$. The concept was originally introduced by Mezei with the first spin echo spectrometer [73] and has since also been applied in other ways [74–77]. Neutron spin echo instruments are, nowadays, operated by almost every major neutron scattering facility [78] and enable important scientific contributions in soft [79] and hard [80] condensed matter research. The state of the art in terms of energy resolution is better than 1 neV with neutrons of 100 μeV energy [81–84].

One recent development that was chosen to be highlighted here is the Magnetic Wollaston Prisms (MWPs) developed at Indiana University (IU) and at ORNL [85–87]. As used in light optics, a Wollaston prism has an inclined interface that separates two calcite prisms with orthogonal optical axes. Such a prism can split unpolarized light into two linearly polarized beams with orthogonal polarization directions. An analogous Wollaston

prism for neutrons has an inclined interface that separates oppositely directed magnetic fields. It can produce a spatial separation between different neutron spin states, which is classically understood as spin precession in an orthogonal field. A neutron passing through the device will acquire a precession phase that is proportional to the inclination angle of the neutron trajectory with a reference direction.

At ORNL, the MWP is constructed using high-temperature superconducting (HTS) materials, including films and tape. The Meissner effect of HTS films is used to confine magnetic fields produced electromagnetically by current-carrying HTS tape wound on triangular soft iron pole pieces, as shown in Figure 3a. With the Meissner effect, the HTS film ensures that the magnetic field transition within the MWP is sharp, well-defined, and planar, which is critical for high-efficiency neutron spin transport.

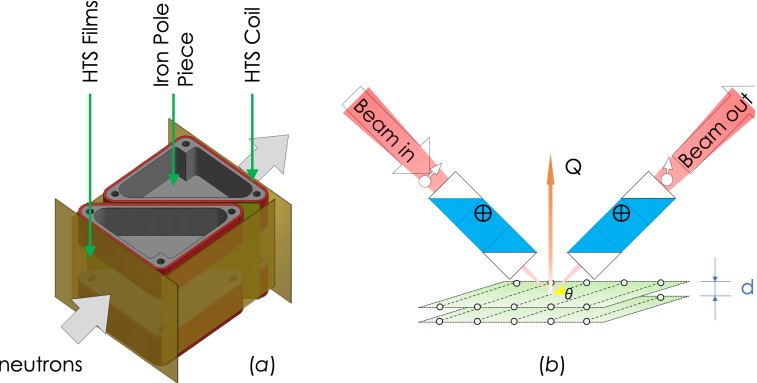

**Figure 3.** (**a**): Neutron Wollaston prism. (**b**): Pair of such prisms with additional rectangular field regions in-between. This particular configuration of two MWPs can measure the *d*-spacing between crystal planes with very high ($\sim 10^{-6}$) relative accuracy.

When the MWP is used alone, it can be used to extend the low limit of the momentum transfer range of a Small-Angle Neutron Scattering (SANS) instrument, via Spin Echo SANS (SESANS), Spin Echo Modulated SANS (SEMSANS), or SANS-SEMSANS. When configured with rectangular magnetic fields, it can be used to improve the momentum transfer resolution or energy resolution of a conventional Triple-Axis Spectrometer, via Inelastic Neutron Spin Echo (INSE) or Larmor Diffraction (LD), as shown in Figure 3b [85–87]. Recently, an MWP has also been used as an entangler–disentangler to achieve mode entanglement between neutron spin and path [88,89].

### 2.5. Data Acquisition Systems

Neutron scattering instruments are long-lived installations, often operating for twenty or more years. Because of the high cost associated with the operation of a neutron source, these are often government-funded research laboratories (ILL) or embedded in one (HFIR and SNS at ORNL). Access to neutron beams is limited and time-limited. Most neutron scattering instruments are operated by user facilities and user access is through a proposal system with peer review. One will get limited time to perform a proposed experiment, and one has to make the best use of the time. This is not on the user alone: the user who uses the neutron beam and the facility that produces the beam both need each other for their success.

A great deal of precious time can be saved with good experiment planning and with computational tools that help to execute an experiment. It is therefore increasingly important to develop such tools and offer them with the instruments to the user community. For example, Q space planning tools for direct geometry spectroscopy are available through the DAVE [90] and MANTID [91] software packages. Another example is The Crystal Plan tool for single crystal diffraction [92]. It takes the available detector space and the amount of Q space to be measured and determines the minimum number of orientations needed to cover that space, thus minimizing over-counting and reducing motions.

As an example, a computational tool is described here that was developed at ORNL and that is referred to as the cuboid tool (see Figure 4). It enables an instrument that maps the local stress in a material, such as NRSF2 at HFIR [93] or VULCAN at SNS [94], to represent an oddly shaped sample in the coordinate system of the instrument, and to vary the volume probed in the scattering while ensuring that this volume stays inside the sample. The tool organizes positional scans like other scans in the data acquisition system and does all the bookkeeping needed to associate the scattering data with metadata and slow controls (motor positions, etc.). The tool can represent any three-dimensional shape that is describable with the .off file format [95].

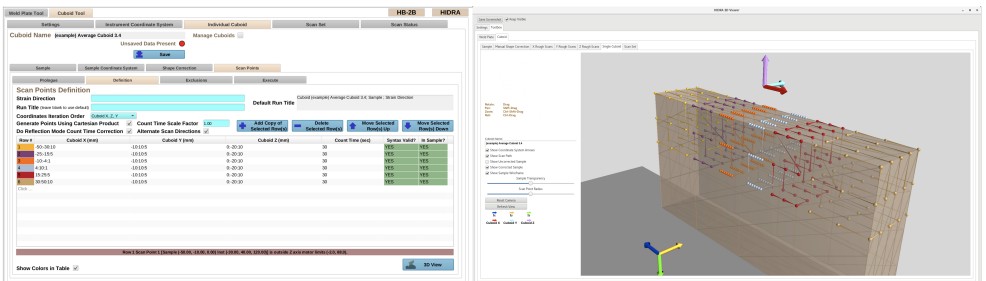

**Figure 4.** Screenshot of the cuboid tool.

Another trend to observe is that instruments are increasingly seen as science laboratories that put a neutron beam on a sample while concurrently performing other measurements or manipulations of the sample [96]. One may also want to extensively prepare a sample offline, next to the instrument, until it is 'ready'. To this end, ORNL has developed mobile carts that can take the data acquisition system (EPICS [97]) to the sample, inclusive of a front-end graphical user interface and server connection.

The first deployment of such a cart was a Mobile EPICS System combined with detector electronics and an eight-pack of $^3$He-tube detectors. This system enabled to take precision measurements of the neutron background in a number of areas at the facility.

The next iteration was a sample environment cart which automated offline sample preparation capability (see Figure 5). This system operates as a standalone sample preparation station which can be wheeled into an instrument area when it is time to conduct the experiment, enabling a seamless transition from sample preparation to experiment start. This cart is in routine use at the Liquids Reflectometer at SNS [98]. This instrument measures specular and off-specular neutron reflectivity in a horizontal sample geometry from solid surfaces, tilted solid/liquid interfaces, or free liquid surfaces. Most of the use of this instrument is in soft matter surface science. Individual component functionalities on the cart are as follows:

- HPLC pump. It provides precise mixing of up to four fluids in a scattering cell. This can be used to change the fluid neutron contrast or fill the cell with the appropriate material to grow a soft matter film in situ. The feature to control flow and maximum pressure is provided.
- Syringe pump. Many film growth processes require an overabundance of material and a continuous exchange with a reservoir. The syringe pump allows users to push the required material back and forth through the cell to provide the optimum film growth conditions while conserving the expensive material. The push–pull sequence can be automatically processed, with controllable volume. The software also provided the control of the syringe dispenser valve positions, pump speed and volume, left/right syringe or both of the syringe pumps together movement, status read-back of pumps and valves, etc.
- Water bath. The water bath provides temperature control so that the beamline users can either hit the optimum condition for film growth or mimic the environment they wish to study the film in.

- Micro-fluid multi-way control valves. By including automated control valves, experimenters can remotely schedule fluid contrast changes or film growth steps between data collection intervals. This minimizes the requirement to enter the instrument and potentially disturb the delicate alignment of the fluid cell. Realigning can take a significant amount of time.

A mobile Linux server on the cart with two network cards connects it to the beamline network and the local facility network. The Input Output Controllers are running on the Linux Operating System on the cart, communicated via local network. Process Variables can be accessed by the beamline server and other Operator Interface software through the Channel Access Gateway.

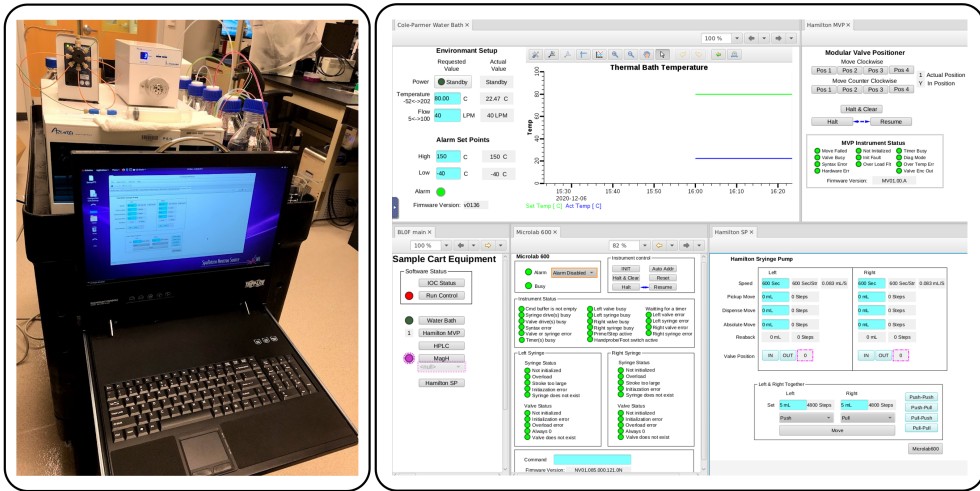

**Figure 5.** Visual of the cart (**left**) and the software interface (**right**).

The Spallation Neutron Source at ORNL was the first facility that consistently collected neutron scattering data with event-based acquisition systems [99,100]. Counting neutrons as events [101], rather than in traditional histograms, enables new types of experiments and also offers various technical advantages.

The breathing of a zeolite lattice during adsorption/desorption of $N_2$ gas was studied with a stroboscopic measurement protocol at the NOMAD instrument [102]. The data were continuously collected under constant flow and pressure but with step-wise varying composition of isotopes $^{14}N_2$ and $^{15}N_2$. Adsorbed $N_2$ contributes to the coherent structure factor, while non-adsorbed (but present) $N_2$ gives rise to diffuse scattering which needs a much longer collection time before the statistical accuracy is sufficient. The two nitrogen isotopes differ in their neutron scattering length and, for this reason, impact the structure factor differently. The data from the different adsorption and desorption steps were re-binned stroboscopically over the measurement cycles to obtain improved statistical accuracy.

Event data enable time-of-flight diffraction measurements in an intense pulsed magnetic field. An example of this type of experiment is the high-field study of field-induced phases in $MnWO_4$ [103]. Because both wavelength and field are time dependent, careful phasing can allow for multiple phases to be measured at the same time. Unusual properties of $LiNiPO_4$ in high magnetic fields, beyond 30 T, including a re-entrant magneto-electric phase, were measured in this way both at the SNS [104] and J-PARC [105].

### 2.6. Virtual Instruments and Experiments Supported by Computer Simulation

Today, it is possible to replicate an instrument in software and to simulate the performance accurately (to within a few %) in a large and multi-dimensional parameter space. The community is actively developing several software packages that employ Monte Carlo methods for the particle transport (McStas [106], MCViNE [107], NCrystal [108], and Vitess [109,110]). This is essential for designing and optimizing new instruments as described in Section 2.1. However, instrument simulation software can also be used to treat

practical problems of neutron scattering science with computational methods that cannot be solved analytically, for example, (i) multiple scattering in the sample, (ii) parasitic scattering from sample environment equipment, or (iii) accurate resolution function calculations, or a combination of these [111,112]. All these effects matter in the sense that they influence the data analysis and the interpretation of the results. They also depend on the sample size and shape, sample scattering law, and instrument configuration. It would be hopeless to attempt to make these calculations analytically.

Sample scattering simulations are becoming more realistic. MCViNE has implemented nested objects and constructive solid geometry that allows multiple scattering kernels and materials to be assembled into complex sample and sample environment simulations [113]. McStas now has similar capabilities with the union components [114]. Furthermore, new tools that allow output in formats for use with CAD software [36] have allowed for streamlined interaction with such software. Specifically, it has accelerated shielding design with MCNP [115]. Another advancement for MCNP is the implementation of rotating frames, which allow for simulating moderation processes in choppers [116,117].

In a few years, a full instrument/experiment simulation will be run in parallel with most if not all scattering experiments, at no additional cost to the user, that exactly represents the chosen instrument configuration in all relevant details. It will be seen as an essential part of the data accumulated during the experiment. The workflow indicated in Figure 6 is one example of how the simulation will support experiments and data analysis. User-friendly software for modeling and the visualization of complex neutron scattering experiments makes it possible to understand the experimental data much better and to develop new analysis protocols which increase the amount of new information gained from experiments significantly.

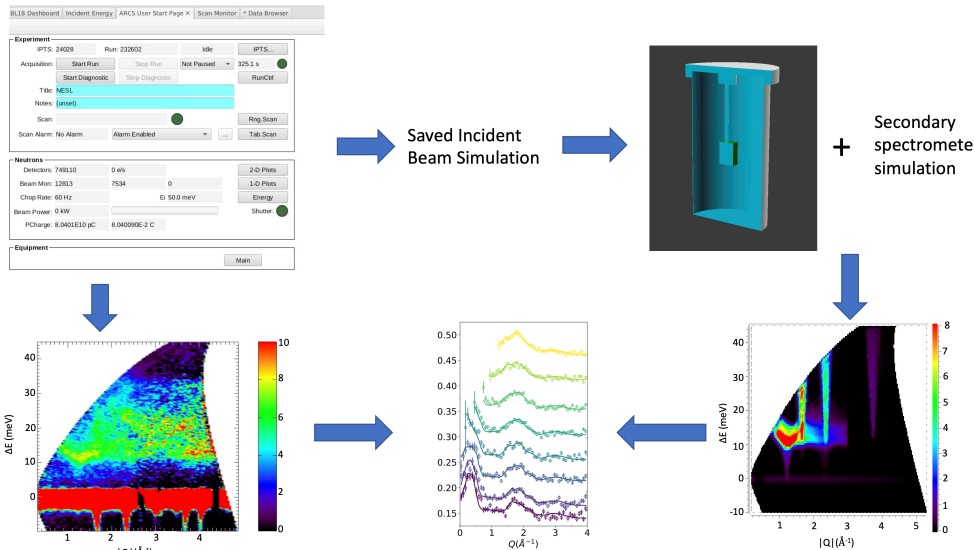

**Figure 6.** Example of a simulation workflow, schematic of what will be performed when the workflow is complete. The simulation of the neutron transport in the beamline is launched at the same time as data acquisition begins and can be refined during the acquisition. After the acquisition, the user combines the simulation and a description of the instrument geometry and scattering cross sections of the sample. The simulations can be plotted and compared to the measurement. The data in the plots are from ref. [118].

The parallel advances in our capabilities to perform computation-heavy molecular dynamics simulations, or lattice dynamics calculations, offer the possibility to integrate such calculations in an experiment workflow where experimental decisions are supported by real-time simulations of the sample scattering [119–121]. Again, this is driven by the desire to maximize the amount of new information gained during an experiment with

limited acquisition time. This would entail the following steps with increasing levels of difficulty and complexity:

1. Model a sample material and calculate dynamical spectra (collective or single-molecule, tunneling, etc.). Nowadays, experiment analysis does not often go beyond this step.
2. Simulate measured spectra, taking into account the instrument geometry and configuration, instrument resolution, and other relevant factors.
3. Simulate spectra during the measurement with the aim to inform experimental decisions in real time.

For the last step, simulations could be specifically geared toward answering questions such as:

1. How can the instrument be set up to bring a feature of interest into the observable range, or to optimally place it in the observable range during an experiment?
2. How long does one need to collect data until a feature of interest can be seen and characterized within a certain desired accuracy?
3. How will a feature of interest evolve with temperature or other parameters?
4. How will a feature of interest evolve if the sample composition is changed?

### 3. Conclusions and Outlook into the Future

The further development of neutron scattering instrumentation is a steady process and will continue to increase the impact of neutron scattering science. Because of the increased cost of technology developments, facilities will probably emphasize areas where the impact of neutron scattering is strong: low-temperature physics, magnetism, and soft matter (which is not to say that other areas will be neglected). New sources and facilities will likely be smaller than the average today, and sources will be designed in parallel with new instruments that are tailored to the needs of the scientific community [122].

**Author Contributions:** G.E. wrote the manuscript with input from all authors. All authors reviewed and contributed to the manuscript. Dome Detector, Y.D. and M.S.L.; instrument design and virtual instruments, G.E.G. and F.F.I.; Wollaston prisms: F.L.; Cart: X.G. and R.O.K.IV; DAQ systems, R.D.G., R.O.K.IV and B.V. All authors have read and agreed to the published version of the manuscript.

**Funding:** This research used resources at the Spallation Neutron Source, a DOE Office of Science User Facility operated by the Oak Ridge National Laboratory. The Dome Detector was developed in a strategic partnership project agreement between ORNL and RMD, Inc., under SPP Agreement No. NFE-21-08574 and SBIR/STTR DOE Award No. DE-SC0020609.

**Institutional Review Board Statement:** Not applicable.

**Informed Consent Statement:** Not applicable.

**Data Availability Statement:** Not applicable.

**Acknowledgments:** The Dome Detector was developed under the leadership of Richard A. Riedel (retired). We acknowledge the support of C. Donahue Jr., C. Montcalm, and T. Visscher for the assembly of the Dome Detector. We also acknowledge M. Waddel's early work on the cart design.

**Conflicts of Interest:** The authors declare no conflict of interest. The funders had no role in the design of the study; in the collection, analyses, or interpretation of data; in the writing of the manuscript; or in the decision to publish the results.

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
