# Peer review of "Modern Trends in Neutron Scattering Instrument Technologies"

_instruments, doi:10.3390/instruments6030022_

Round 1

Author Response

Modern Trends in Neutron Scattering Instrument Technologies

By G. Ehlers et al.

Submitted to Instruments (2022)

Manuscript ID: instruments-1813303

Author’s response to comments by reviewer 1

On behalf of all authors, I would like to thank the reviewer for the critical, yet positive and supportive, review.  The reviewer notes that “The paper is absolutely suited for publication in the MDPI journal Instruments”.  Our responses to the specific points raised by the reviewer are as follows.

Line 18. The reviewer is of course right in pointing out that the FRM-II reactor deserves to be mentioned in the same sentence with ILL and HFIR as a high-brightness source. The sentence has been modified and a proper reference was added.

Line 75. Other facilities are perhaps/probably ahead of ORNL in this development. The sentence was rephrased to remove the impression that ORNL was leading this effort.

Line 100. This sentence was changed accordingly, and it now includes a reference to the experimental resolution which will in general become coarser when the beam divergence is increased.

Line 218. The article suggested by the reviewer is certainly a meaningful reference in the context here and the citation was added after “…improve the count rate capability and the spatial resolution.”

Georg Ehlers

Neutron Instrument Technologies Section

Neutron Technologies Division

Oak Ridge National Laboratory

1 Bethel Valley Road, Oak Ridge, TN 37831-6466

Reviewer 2 Report

The manuscript presents a review about new developments in the field of neutron scattering technology. It gives an overview about recent trends and provides an outlook where the journey might go. It focuses on developments in the US and is a highly welcome contribution demonstrating that the US is now also developing new technologies in the field of neutron scattering techniques. The article is well written and provides a good insight into new ideas to improve neutron scattering instrumentation.

I recommend the publication of this manuscript after two minor points are clarified:

1)     “Brightness” is used a few times, but in the definition at least “per unit time” is missing. May be the authors wanted to use “brilliance” instead? This point needs clarification and a more precise definition.

2)     In fig 2 it is practically impossible to “see” the diffraction pattern.

Author Response

Modern Trends in Neutron Scattering Instrument Technologies

By G. Ehlers et al.

Submitted to Instruments (2022)

Manuscript ID: instruments-1813303

Author’s response to comments by reviewer 2

On behalf of all authors, I would like to thank the reviewer for the critical, yet positive and supportive, review.  We note that the reviewer also recommends the publication of our article.  Our responses to the specific points raised by the reviewer are as follows.

  1. We would like to thank the reviewer for pointing out that our brightness definition was incorrect. This was clearly an oversight on our part and has been corrected (unit time has been added to the brightness definition). 
  2. Regarding figure 2, the displayed image on the right-hand side is not meant to represent an entire diffraction pattern, but only two Bragg peaks that were simultaneously collected in a demonstration. The figure caption was misleading in this regard and has been modified.

Georg Ehlers

Neutron Instrument Technologies Section

Neutron Technologies Division

Oak Ridge National Laboratory

1 Bethel Valley Road, Oak Ridge, TN 37831-6466
